# Batch-Size Independent Regret Bounds for Combinatorial Semi-Bandits with Probabilistically Triggered Arms or Independent Arms

**Xutong Liu**
The Chinese University of Hong Kong
Hong Kong SAR, China
liuxt@cse.cuhk.edu.hk

**Jinhang Zuo**
Carnegie Mellon University
Pittsburgh, PA, USA
jzuo@andrew.cmu.edu

**Siwei Wang**
Microsoft Research
Beijing, China
siweiwang@microsoft.com

**Carlee Joe-Wong**
Carnegie Mellon University
Pittsburgh, PA, USA
cjoewong@andrew.cmu.edu

**John C.S. Lui**
The Chinese University of Hong Kong
Hong Kong SAR, China
cslui@cse.cuhk.edu.hk

**Wei Chen**
Microsoft Research
Beijing, China
weic@microsoft.com

## Abstract

In this paper, we study the combinatorial semi-bandits (CMAB) and focus on reducing the dependency of the batch-size $K$ in the regret bound, where $K$ is the total number of arms that can be pulled or triggered in each round. First, for the setting of CMAB with probabilistically triggered arms (CMAB-T), we discover a novel (directional) triggering probability and variance modulated (TPVM) condition that can replace the previously-used smoothness condition for various applications, such as cascading bandits, online network exploration and online influence maximization. Under this new condition, we propose a BCUCB-T algorithm with variance-aware confidence intervals and conduct regret analysis which reduces the $O(K)$ factor to $O(\log K)$ or $O(\log^2 K)$ in the regret bound, significantly improving the regret bounds for the above applications. Second, for the setting of non-triggering CMAB with independent arms, we propose a SESCB algorithm which leverages on the non-triggering version of the TPVM condition and completely removes the dependency on $K$ in the leading regret. As a valuable by-product, the regret analysis used in this paper can improve several existing results by a factor of $O(\log K)$. Finally, experimental evaluations show our superior performance compared with benchmark algorithms in different applications.

## 1 Introduction

Stochastic multi-armed bandit (MAB) [26, 3, 4] is a classical model that has been extensively studied in online decision making. As an extension of MAB, combinatorial multi-armed bandits (CMAB) have drawn much attention recently, owing to its wide applications in marketing, network optimization and online advertising [13, 17, 7, 8, 29, 23]. In CMAB, the learning agent chooses a combinatorial action in each round, and this action would trigger a set of arms (or a super arm) to be pulled simultaneously, and the outcomes of these pulled arms are observed as feedback. Typically, such feedback is known as the semi-bandit feedback. The agent's goal is to minimize the expected *regret*, which is the difference in expectation for the overall rewards between always playing the best action

36th Conference on Neural Information Processing Systems (NeurIPS 2022).

Table 1: Summary of the algorithms and results for CMAB with probabilistically triggered arms.

| Algorithm | Smoothness | Independent Arms? | Computation | Regret |
|---|---|---|---|---|
| CUCB [29] | 1-norm TPM, $B_1$ | Not required | Efficient | $O(K \sum_{i \in [m]} \frac{B_1^2 \log T}{\Delta_i^{\min}})$ |
| BOIM-CUCB [25, Section 4]* | 1-norm TPM, $B_1$ | Required | Hard | $O((\log K)^2 \sum_{i \in [m]} \frac{B_1^2 \log T}{\Delta_i^{\min}})$ |
| BCUCB-T (Algorithm 1) | TPVM$_<$, $B_v$,† $\lambda > 1$ | Not required | Efficient | $O(\log K \sum_{i \in [m]} \frac{B_v^2 \log T}{\Delta_i^{\min}})$ |
| BCUCB-T (Algorithm 1) | TPVM$_<$, $B_v$,† $\lambda = 1$ | Not required | Efficient | $O((\log K \log \frac{B_v K}{\Delta_{\min}} \sum_{i \in [m]} \frac{B_v^2 \log T}{\Delta_i^{\min}})$ |
| BOIM-CUCB (Appendix C)‡ | 1-norm TPM, $B_1$ | Required | Hard | $O(\log K \sum_{i \in [m]} \frac{B_1^2 \log T}{\Delta_i^{\min}})$ |

* This work is for a specific application, but we treat it as a general framework;  † Generally, $B_v = O(B_1 \sqrt{K})$, and the existing regret bound is improved when $B_v = o(B_1 \sqrt{K})$;  ‡ Using our new analysis.

(i.e., the action with highest expected reward) and playing according to the agent's own policy. For CMAB, an agent not only need to deal with the exploration-exploitation tradeoff: whether the agent should explore arms in search for a better action, or should the agent stick to the best action observed so far to gain rewards; but also need to handle the exponential explosion of all possible actions.

To model a wider range of application scenarios where action may trigger arms probabilistically, Chen et al. [8] first generalize CMAB to CMAB with probabilistically triggered arms (or CMAB-T for short), which successfully covers cascading bandit [9] (CB) and online influence maximization (OIM) bandit [31] problems. Later on, Wang and Chen [29] improve the regret bound of [8] by introducing a smoothness condition, called the triggering probability modulated (TPM) condition, which removes a factor of $1/p^*$ compared to [8], where $p^*$ is the minimum positive probability that any arm can be triggered. However, in both studies, the regret bounds still depend on a factor of *batch-size $K$*, where $K$ is the maximum number of arms that can be triggered, and this factor could be quite large, e.g., for OIM $K$ can be as large as the number of edges in a large social network.

**Our Contributions.** In this paper, we reduce or remove the dependency on $K$ in the regret bounds. For CMAB-T, we first discover a new *triggering probability and variance modulated (TPVM)* bounded smoothness condition, which is stronger than the TPM condition, yet still holds for several applications (such as CB and OIM) where only the TPM condition is known previously. We observe that for these applications, the previous TPM condition bounds the global speed of reward change regarding the parameter change, which will cause a large $K$ coefficient due to the rapid change at the boundary regions (i.e., when an arm's mean $\mu_i$ is close to 0 or 1). Our TPVM condition utilizes this observation by raising up the regret contribution of those boundary regions, leading to a significant reduction on the dependency of $K$. Second, we propose a "variance-aware" BCUCB-T algorithm that adaptively changes the width of the confidence interval according to the (empirical) variance, cancelling out the large regret contribution raised by the TPVM condition at the boundary regions (where the variances are also very small). Combining these two techniques, we successfully reduce the batch-size dependence from $O(K)$ to $O(\log K)$ or $O(\log^2 K)$ for all CMAB-T problems satisfying the TPVM condition, leading to significant improvements of the regret bounds for applications like CB or OIM. As a by-product, we also give refined proofs that shall improve the regret for several existing works by a factor of $O(\log K)$, e.g., [11, 23], which may be of independent interests.

In addition to the general CMAB-T setting, we show how a non-triggering version of the TPVM condition (i.e., VM condition) can help to completely remove the batch-size $K$, under the additional independent arm assumption for non-triggering CMAB problems. In particular, we propose a novel Sub-Exponential Efficient Sampling for Combintorial Bandits Policy (SESCB) that produces tighter sub-exponential concentrated confidence intervals. In our analysis, we show that the total regret only depends on the arm that is observed least instead of all $K$ arms, so that we can achieve a completely batch-size independent regret bound. Our empirical results demonstrate that our proposed algorithms can achieve around 20% lower regrets than previous ones for several applications. Due to the space limit, we will move the complete proofs and empirical results into the appendix.

**Related Work.** The stochastic CMAB has received much attention recently. From the modelling point of view, these CMAB works can be divided into two categories: CMAB with or without probabilistically triggered arms (i.e. CMAB-T setting or non-triggering CMAB). For CMAB-T, our work improves (a) the general framework in [8, 29], (b) the combinatorial cascading bandit [17], (c) the online multi-layered network exploration [21] problem, (d) the online influence maximization bandits [29, 25], by reducing or removing the batch-size dependent factor $K$ in the regret bounds with our new TPVM condition and/or our refined analysis. We defer the detailed technical comparison to Section 3.1 and Section 5. For the algorithm, most CMAB-T studies use Combinatorial Upper Confidence Bound (CUCB) based on Chernoff concentration bounds [29], our BCUCB-T algorithm is different and uses the Bernstein concentration bound [2, 23] that considers variance of the arms.

Table 2: Summary of the algorithms and results for non-triggering CMAB problems.

| Algorithm | Smoothness | Independent Arms? | Computation | Regret |
|---|---|---|---|---|
| CUCB [29] | 1-norm, $B_1$ | Not required | Efficient | $O(K\sum_{i\in[m]}\frac{B_1^2\log T}{\Delta_i^{\min}})$ |
| CTS [30]* | 1-norm, $B_1$ | Required | Efficient | $O(K\sum_{i\in[m]}\frac{B_1^2\log T}{\Delta_i^{\min}})$ |
| ESCB [9] | 1-norm, $B_1$** | Required | Hard | $O((\log K)^2\sum_{i\in[m]}\frac{B_1^2\log T}{\Delta_i^{\min}})$ |
| AESCB [10] | Linear | Required | Efficient | $O((\log K)^2\sum_{i\in[m]}\frac{\log T}{\Delta_i^{\min}})$ |
| BC-UCB [23]† | VM, $B_v$‡ | Not required | Efficient | $O((\log K)^2\sum_{i\in[m]}\frac{B_v^2\log T}{\Delta_i^{\min}})$. |
| CTS [30]* | Linear | Required | Efficient | $O(\log K\sum_{i\in[m]}\frac{\log T}{\Delta_i^{\min}})$ |
| SESCB (Algorithm 2) | VM, $B_v$‡ | Required | Efficient*** | $O(\sum_{i\in[m]}\frac{B_v^2\log T}{\Delta_i^{\min}})$ |
| BC-UCB (Appendix C)§ | VM, $B_v$‡ | Not required | Efficient | $O(\log K\sum_{i\in[m]}\frac{B_v^2\log T}{\Delta_i^{\min}})$ |

* Requires exact offline oracle instead of $(\alpha,\beta)$-approximate oracle;   † This work gives sufficient smoothness condition with factor $\gamma_g$ and translates to $B_v=3\sqrt{2}\gamma_g$ in our setting;   § Using our new analysis.   ** This work is for the linear case, but can easily generalize to 1-norm $B_1$ case;   ‡ Generally, $B_v=O(B_1\sqrt{K})$ and the existing regret bound is improved when $B_v=o(B_1\sqrt{K})$; *** Efficient when the reward function is submodular, otherwise the computation is hard;

For non-triggering CMAB, [13] is the first study on stochastic CMAB, and its regret has been improved by Kveton et al. [18], Combes et al. [9], Chen et al. [8], but they still have $O(K)$ factor in their regrets. When arms are mutually independent, Combes et al. [9] build a tighter ellipsoidal confidence region for exploration, and devise the Efficient Sampling for Combinatorial Bandit policy (ESCB), which reduces the dependence on $O(K)$ to $O(\log^2 K)$ at the cost of high computational complexity (since combinatorial optimization over the ellipsoidal region is NP-hard in general [1]). Later on, the computational complexity is improved by AESCB [10] in the linear CMAB problem. Recently, Merlis and Mannor [23] focus on the Probabilistic Maximum Coverage (PMC) bandit problem and propose the BC-UCB algorithm with the Gini-smoothness condition to achieve a similar improvement as ESCB/AESCB, but without the independent arm assumption. Our work is largely inspired by their work, however, our study generalizes theirs to the CMAB-T setting which can handle much broader application scenarios beyond the non-triggering CMAB (more detailed comparison is given in Section 3). In addition, we provide a refined analysis that can save a $O(\log K)$ factor for BC-UCB (or ESCB/AESCB) algorithm. Compared with other ESCB-type algorithms for independent arms, as far as we know, our SESCB algorithm are the first to completely remove the dependence of $K$ in the leading regrets, owing to our non-triggering version of the TPVM condition. The detailed comparisons are summarized in Table 1 and Table 2.

The usage of variance-aware algorithms to give improved regret bounds can be dated back to [2]. Recently, there is a surge of interest to apply the variance-aware principle in bandit [23, 28] and reinforcement learning (RL) settings [33, 32]. It is notable that Vial et al. [28] share a similar variance-aware principle as ours but focus on the distribution-independent regret bounds for the cascading bandits [28]. Our work is more general and achieves the matching regret bound when translating to the distribution-independent regret bound. Compared with RL works, our paper studies a different setting as we do not consider the state transitions.

From the application's point of view, this paper covers the applications of PMC bandit [23], combinatorial cascading bandits [17, 19], network exploration [21], and online influence maximization [31, 29, 20]. Our proposed algorithms can significantly reduce the regret bounds of them, e.g., from $O(K)$ to $O(\log^2 K)$ for OIM where $K$ can be hundreds of thousands in large social networks.

## 2  Problem Settings

We study the combinatorial multi-armed bandit problem with probabilistic triggering arms, which is denoted as CMAB-T for short. Following the setting from [29], a CMAB-T *problem instance* can be described by a tuple $([m], \mathcal{S}, \mathcal{D}, D_{\text{trig}}, R)$, where $[m] = \{1, 2, ..., m\}$ is the set of base arms; $\mathcal{S}$ is the set of eligible actions and $S \in \mathcal{S}$ is an action;[1] $\mathcal{D}$ is the set of possible distributions over the outcomes of base arms with bounded support $[0, 1]^m$; $D_{\text{trig}}$ is the probabilistic triggering function and $R$ is the reward function, the definitions of which will be introduced shortly.

In CMAB-T, the learning agent interacts with the unknown environment in a sequential manner as follows. First, the environment chooses a distribution $D \in \mathcal{D}$ unknown to the agent. Then, at round

---

[1]In some cases $\mathcal{S}$ is a collection of subsets of $[m]$, in which case we often refer to $S \in \mathcal{S}$ as a super arm. In this paper we treat $\mathcal{S}$ as a general action space, same as in [29].

$t = 1, 2, ..., T$, the agent selects an action $S_t \in \mathcal{S}$ and the environment draws from the unknown distribution $D$ a random outcome $\boldsymbol{X}_t = (X_{t,1}, ...X_{t,m}) \in [0,1]^m$. Note that the outcome $\boldsymbol{X}_t$ is assumed to be independent from outcomes generated in previous rounds, but outcomes $X_{t,i}$ and $X_{t,j}$ in the same round could be correlated. Let $D_{\text{trig}}(S, \boldsymbol{X})$ be a distribution over all possible subsets of $[m]$, i.e. its support is $2^{[m]}$. When the action $S_t$ is played on the outcome $\boldsymbol{X}_t$, base arms in a random set $\tau_t \sim D_{\text{trig}}(S_t, \boldsymbol{X}_t)$ are triggered, meaning that the outcomes of arms in $\tau_t$, i.e. $(X_t)_{t \in \tau_t}$ are revealed as the feedback to the agent, and are involved in determining the reward of action $S_t$. Function $D_{\text{trig}}$ is referred as the *probabilistic triggering function*. At the end of the round $t$, the agent will receive a non-negative reward $R(S_t, \boldsymbol{X}_t, \tau_t)$, determined by $S_t, \boldsymbol{X}_t$ and $\tau_t$. CMAB-T significantly enhances the modeling power of CMAB [7, 18] and can model many applications such as cascading bandits and online influence maximization [29], which we will discuss in later sections.

The goal of CMAB-T is to accumulate as much reward as possible over $T$ rounds, by learning distribution $D$ or its parameters. Let $\boldsymbol{\mu} = (\mu_1, ..., \mu_m)$ denote the mean vector of base arms' outcomes. Following [29], we assume that the expected reward $\mathbb{E}[R(S, \boldsymbol{X}, \tau)]$ is a function of the unknown mean vector $\boldsymbol{\mu}$, where the expectation is taken over the randomness of $\boldsymbol{X} \sim D$ and $\tau \sim D_{\text{trig}}(S, \boldsymbol{X})$. In this context, we denote $r(S; \boldsymbol{\mu}) \triangleq \mathbb{E}[R(S, \boldsymbol{X}, \tau)]$ and it suffices to learn the unknown mean vector instead of the joint distribution $D$, based on the past observation.

The performance of an online learning algorithm $A$ is measured by its *regret*, defined as the difference of the expected cumulative reward between always playing the best action $S^* \triangleq \text{argmax}_{S \in \mathcal{S}} r(S; \boldsymbol{\mu})$ and playing actions chosen by algorithm $A$. For many reward functions, it is NP-hard to compute the exact $S^*$ even when $\boldsymbol{\mu}$ is known, so similar to [29], we assume that the algorithm $A$ has access to an offline $(\alpha, \beta)$-approximation oracle, which for mean vector $\boldsymbol{\mu}$ outputs an action $S$ such that $\Pr[r(S; \boldsymbol{\mu}) \geq \alpha \cdot r(S^*; \boldsymbol{\mu})] \geq \beta$. Formally, the $T$-round $(\alpha, \beta)$-approximate regret is defined as

$$Reg(T; \alpha, \beta, \boldsymbol{\mu}) = T \cdot \alpha\beta \cdot r(S^*; \boldsymbol{\mu}) - \mathbb{E}\left[\sum_{t=1}^{T} r(S_t; \boldsymbol{\mu})\right], \tag{1}$$

where the expectation is taken over the randomness of outcomes $\boldsymbol{X}_1, ..., \boldsymbol{X}_T$, the triggered sets $\tau_1, ..., \tau_T$, as well as the randomness of algorithm $A$ itself.

In the CMAB-T model, there are several quantities that are crucial to the subsequent study. We define *triggering probability* $p_i^{D, D_{\text{trig}}, S}$ as the probability that base arm $i$ is triggered when the action is $S$, the outcome distribution is $D$, and the probabilistic triggering function is $D_{\text{trig}}$. Since $D_{\text{trig}}$ is always fixed in a given application context, we ignore it in the notation for simplicity, and use $p_i^{D, S}$ henceforth. Triggering probabilities $p_i^{D, S}$'s are crucial for the triggering probability modulated bounded smoothness conditions to be defined below. We define *batch size* $K$ as the maximum number of arms that can be triggered, i.e., $K = \max_{S \in \mathcal{S}} |\{i \in [m] : p_i^{D, S} > 0\}|$. Our main contribution of this paper is to remove or reduce the regret dependency on batch size $K$, where $K$ could be quite large, e.g., $K$ can be hundreds of thousands in a large social network.

Owing to the nonlinearity and the combinatorial structure of the reward, it is essential to give some conditions for the reward function in order to achieve any meaningful regret bounds [7, 8, 29, 11, 23]. The following are two standard conditions originally proposed by Wang and Chen [29].

**Condition 1** (Monotonicity). *We say that a CMAB-T problem instance satisfies monotonicity condition, if for any action $S \in \mathcal{S}$, any two distributions $D, D' \in \mathcal{D}$ with mean vectors $\boldsymbol{\mu}, \boldsymbol{\mu}' \in [0,1]^m$ such that $\mu_i \leq \mu_i'$ for all $i \in [m]$, we have $r(S; \boldsymbol{\mu}) \leq r(S; \boldsymbol{\mu}')$.*

**Condition 2** (1-norm TPM Bounded Smoothness). *We say that a CMAB-T problem instance satisfies the triggering probability modulated (TPM) $B_1$-bounded smoothness condition, if for any action $S \in \mathcal{S}$, any distribution $D, D' \in \mathcal{D}$ with mean vectors $\boldsymbol{\mu}, \boldsymbol{\mu}' \in [0,1]^m$, we have $|r(S; \boldsymbol{\mu}') - r(S; \boldsymbol{\mu})| \leq B_1 \sum_{i \in [m]} p_i^{D,S} |\mu_i - \mu_i'|$.*

The first monotonicity condition indicates the reward is larger if the parameter vector $\boldsymbol{\mu}$ is larger. The second condition bounds the reward difference caused by the parameter change (from $\boldsymbol{\mu}$ to $\boldsymbol{\mu}'$). One key feature is that the parameter change in each base arm $i \in [m]$ is modulated by the triggering probability $p_i^{D,S}$. Intuitively, for base arm $i$ that is unlikely to be triggered/observed (small $p_i^{D,S}$), Condition 2 ensures that a large change in $\mu_i$ only causes a small change (multiplied by $p_i^{D,S}$) in the reward, and thus one does not need to pay extra cost to observe such arms. Many applications satisfy Condition 1 and Condition 2, including linear combinatorial bandits [18], combinatorial

cascading bandits [17], online influence maximization [29], etc. With the above two conditions, Wang and Chen [29] show that a CUCB algorithm achieves the distribution-dependent regret bound of $O(\sum_{i \in [m]} \frac{B_1^2 K \log T}{\Delta_i^{\min}})$, where $\Delta_i^{\min}$ is the distribution-dependent reward gap, to be formally defined in Definition 1. In the following sections, we will show how to remove or reduce the dependency on $K$ in the above bounds under our new conditions.

# 3  Algorithm and Regret Analysis for CMAB-T

In this section, for the CMAB-T framework with probabilistic triggering, we improve the regret dependency on the batch size from $O(K)$ in [29] to $O(\log K)$ or $O(\log^2 K)$. Our main tool is a new condition called *triggering probability and variance modulated (TPVM) bounded smoothness condition*, replacing the TPM condition (Condition 2). We will define the TPVM condition, comparing it with the TPM condition and the gini-smoothness condition of [23], show our algorithm and regret analysis that utilize this condition. Later in Section 5, we will demonstrate how this condition is applied to applications such as cascading bandits and online influence maximization.

## 3.1  Triggering Probability and Variance Modulated (TPVM) Bounded Smoothness Condition

In this paper, we discover a new smoothness condition for many important applications as follows.

**Condition 3** (Directional TPVM Bounded Smoothness). *We say that a CMAB-T problem instance satisfies the directional TPVM $(B_v, B_1, \lambda)$-bounded smoothness condition $(B_v, B_1 \geq 0, \lambda \geq 1)$, if for any action $S \in \mathcal{S}$, any distribution $D, D' \in \mathcal{D}$ with mean vector $\boldsymbol{\mu}, \boldsymbol{\mu}' \in (0,1)^m$, for any non-negative $\boldsymbol{\zeta}, \boldsymbol{\eta} \in [0,1]^m$ s.t. $\boldsymbol{\mu}' = \boldsymbol{\mu} + \boldsymbol{\zeta} + \boldsymbol{\eta}$, we have*

$$|r(S; \boldsymbol{\mu}') - r(S; \boldsymbol{\mu})| \leq B_v \sqrt{\sum_{i \in [m]} (p_i^{D,S})^\lambda \frac{\zeta_i^2}{(1 - \mu_i)\mu_i}} + B_1 \sum_{i \in [m]} p_i^{D,S} \eta_i. \tag{2}$$

**Remark 1 (Intuition for Condition 3).** Looking at Eq. (2), if we ignore the $(1 - \mu_i)\mu_i$ term in the denominator and set $\lambda = 2$, the RHS of Eq. (2) becomes $B_v \sqrt{\sum_{i \in [m]} (p_i^{D,S})^2 \zeta_i^2} + B_1 \sum_{i \in [m]} p_i^{D,S} \eta_i$, which holds with $B_v = B_1 \sqrt{K}$ by applying the Cauchy-Schwarz inequality to Condition 2. However, the regret upper bound following this modified Eq. (2) would not directly lead to the improvement in the regret due to the $\sqrt{K}$ factor in $B_v$. To deal with this issue, an important observation here is that for many applications, the reason $B_v$ is large is because that the reward changes abruptly when parameters $\mu_i$ approaches 0 or 1. This motivates us to plug in the $1/(1 - \mu_i)\mu_i$ term in Eq. (2) to enlarge the square root term when $\mu_i$ is close to 0 or 1, so that $B_v$ can be as small as possible. On the other hand, notice that when $\mu_i$ approaches 0 or 1, the variance $V_i \leq (1 - \mu_i)\mu_i$ is also very small, [2] so the estimation of $\mu_i$ should be quite accurate. Therefore, the gap $\zeta_i$ between our estimation and true value produces a variance-related term which cancels the $(1 - \mu_i)\mu_i$ in the denominator. Since $\zeta_i$ in Eq. (2) is modulated by both triggering probability $p_i^{D,S}$ and inverse upper bound of the variance $1/(1 - \mu_i)\mu_i$, we call Condition 3 the directional triggering probability and variance modulated (TPVM) condition for short, where the term "directional" is explained in the next remark. The exponent $\lambda \geq 1$ on the triggering probability gives flexibility to trade-off between the strength of the condition and the quantity of the regret bound: With a larger $\lambda$, we can obtain a smaller regret bound, while with a smaller $\lambda$, the condition is easier to satisfy and allows us to include more applications.

**Remark 2 (On directional TPVM vs. undirectional TPVM).** In the above definition, "directional" means that we have $\boldsymbol{\zeta}, \boldsymbol{\eta} \geq \mathbf{0}$ such that $\boldsymbol{\mu}' \geq \boldsymbol{\mu}$ in every dimension. This is weaker than the version of the undirectional TPVM condition, where $\boldsymbol{\zeta}, \boldsymbol{\eta} \in [-1,1]^m$, and the $\eta_i$ in the right hand side of Eq.(2) is replaced with $|\eta_i|$. The reason we use the weaker version is that some of our applications considered in this paper only satisfy the weaker version. To differentiate, we use TPVM$_<$ when we refer to the directional TPVM condition.

**Remark 3 (Relation between Conditions 2 and 3).** First, when setting $\boldsymbol{\zeta}$ to $\mathbf{0}$, the directional TPVM condition degenerates to the directional TPM condition. However, Condition 2 is the undirectional TPM condition, which is typically stronger than its directional counterpart. Thus, in general Condition 3 does not imply Condition 2. Nevertheless, with some additional assumptions Condition 3 does imply

---

[2] For bounded random variable $X \in [0,1]$ with mean $\mu_i$, variance $V_i = \mathbb{E}[X^2] - \mathbb{E}[X]^2 \leq \mathbb{E}[X] - (\mathbb{E}[X])^2 \leq (1 - \mu_i)\mu_i$, where the equality is achieved when $X$ is a Bernoulli random variable.

---

**Algorithm 1** BCUCB-T: Bernstein Combinatorial Upper Confidence Bound Algorithm for CMAB-T

1: **Input:** Base arms $[m]$, computation oracle ORACLE.
2: **Initialize:** For each arm $i$, $T_{0,i} \leftarrow 0, \hat{\mu}_{0,i} = 0, \hat{V}_{0,i} = 0$.
3: **for** $t = 1, ..., T$ **do**
4:     For arm $i$, compute $\rho_{t,i}$ according to Eq. (3) and set UCB value $\bar{\mu}_{t,i} = \min\{\hat{\mu}_{t-1,i} + \rho_{t,i}, 1\}$.
5:     $S_t = \text{ORACLE}(\bar{\mu}_{t,1}, ..., \bar{\mu}_{t,m})$.
6:     Play $S_t$, which triggers arms $\tau_t \subseteq [m]$ with outcome $X_{t,i}$'s, for $i \in \tau_t$.
7:     For every $i \in \tau_t$, update $T_{t,i} = T_{t-1,i} + 1$, $\hat{\mu}_{t,i} = \hat{\mu}_{t-1,i} + (X_{t,i} - \hat{\mu}_{t-1,i})/T_{t,i}$, $\hat{V}_{t,i} = \frac{T_{t-1,i}}{T_{t,i}}\left(\hat{V}_{t-1,i} + \frac{1}{T_{t,i}}\left(\hat{\mu}_{t-1,i} - X_{t,i}\right)^2\right)$.
8: **end for**

---

Condition 2 with the same coefficient $B_1$ (See Appendix A for an example of such assumptions). Conversely, by applying the Cauchy-Schwartz inequality, one can verify that if a reward function is TPM $B_1$-bounded smooth, then it is (directional) TPVM $(B_1\sqrt{K}/2, B_1, \lambda)$-bounded smooth for any $\lambda \leq 2$. For applications considered in this paper, we are able to reduce their $B_v$ coefficient from $B_1\sqrt{K}/2$ to a coefficient independent of $K$, leading to significant savings in the regret bound.

**Remark 4 (Comparing with [23]).** Merlis and Mannor [23] introduce a Gini-smoothness condition to reduce the batch-size dependency for CMAB problems, which largely inspires our TPVM$_<$ condition. Their condition is specified in a differential form of the reward function, with parameters $\gamma_\infty$ and $\gamma_g$ (See Appendix B for the exact definition). We emphasize that their original condition cannot handle the probabilistic triggering setting in CMAB-T. One natural extension is to incorporate triggering probability modulation into their differential form of Gini-smoothness. However, we found that the resulting TPM Gini-smoothness condition is not strong enough to guarantee desirable regret bounds (See Appendix B.1). This motivates us to provide a new condition directly on the difference form $|r(S; \boldsymbol{\mu}') - r(S; \boldsymbol{\mu})|$, similar to the TPM condition in [29]. Our TPVM$_<$ condition (Condition 3) can be viewed as extending Lemma 6 of [23] to incorporate triggering probabilities and bound the difference form $|r(S; \boldsymbol{\mu}') - r(S; \boldsymbol{\mu})|$. Intuitively, $B_1$ and $B_v$ correspond to $\gamma_\infty$ and $\gamma_g$, respectively, but since they are for different forms of definitions, their numerical values may not exactly match one another.

### 3.2 BCUCB-T Algorithm and Regret Analysis

Our proposed algorithm BCUCB-T is a generalization of the BC-UCB algorithm [23, Algorithm 1] which originally solves the non-triggering CMAB problem. Algorithm 1 maintains the empirical estimate $\hat{\mu}_{t,i}$ and $\hat{V}_{t,i}$ for the true mean and the true variance of the base arm outcomes. To select the action $S_t$, it feeds the upper confidence bound $\bar{\mu}_i$ into the offline oracle, where $\bar{\mu}_i$ optimistically estimates the $\mu_i$ by a confidence interval $\rho_{t,i}$. Compared with the CUCB algorithm [29, Algorithm 1] which uses confidence interval $\rho_{t,i} = \sqrt{\frac{3\log t}{2T_{t-1,i}}}$ for the CMAB-T problem, the novel part is the usage of empirical variance $\hat{V}_{t-1,i}$ to construct the following "variance-aware" confidence interval:

$$\rho_{t,i} = \sqrt{\frac{6\hat{V}_{t-1,i}\log t}{T_{t-1,i}}} + \frac{9\log t}{T_{t-1,i}} \tag{3}$$

This confidence interval leverages on the empirical Bernstein inequality instead of the Chernoff-Hoeffding inequality. As we will show in Appendix C.1, for the first term in Eq. (3), $\hat{V}_{t-1,i}$ is approximately equal to the true variance $V_i \leq (1 - \mu_i)\mu_i$ and this indicates the estimation of $\mu_i$ is more accurate when $\mu_i$ is close to 0 or 1, which will cancel out the $(1 - \mu_i)\mu_i$ coefficient of the $B_v$ term in Condition 3 as we discussed before. The second term of Eq. (3) is to compensate the usage of the empirical variance $\hat{V}_{t-1,i}$, rather than the true variance $V_i$ which is unknown to the learner.

To state the regret bound, we first give some definitions followed by our main result.

**Definition 1** ((Approximation) Gap). *Fix a distribution $D \in \mathcal{D}$ and its mean vector $\boldsymbol{\mu}$, for each action $S \in \mathcal{S}$, we define the (approximation) gap as $\Delta_S = \max\{0, \alpha r(S^*; \boldsymbol{\mu}) - r(S; \boldsymbol{\mu})\}$. For each arm $i$, we define $\Delta_i^{\min} = \inf_{S \in \mathcal{S}: p_i^{D,S} > 0, \Delta_S > 0} \Delta_S$, $\Delta_i^{\max} = \sup_{S \in \mathcal{S}: p_i^{D,S} > 0, \Delta_S > 0} \Delta_S$. As a convention, if there is no action $S \in \mathcal{S}$ such that $p_i^{D,S} > 0$ and $\Delta_S > 0$, then $\Delta_i^{\min} = +\infty, \Delta_i^{\max} = 0$. We define $\Delta_{\min} = \min_{i \in [m]} \Delta_i^{\min}$ and $\Delta_{\max} = \max_{i \in [m]} \Delta_i^{\min}$.*

**Theorem 1.** *For a CMAB-T problem instance $([m], \mathcal{S}, \mathcal{D}, D_{trig}, R)$ that satisfies monotonicity (Condition 1), and $\text{TPVM}_<$ bounded smoothness (Condition 3) with coefficient $(B_v, B_1, \lambda)$,*

*(1) if $\lambda > 1$, BCUCB-T (Algorithm 1) with an $(\alpha, \beta)$-approximation oracle achieves an $(\alpha, \beta)$-approximate regret bounded by*

$$O\left(\sum_{i \in [m]} \frac{B_v^2 \log K \log T}{\Delta_i^{\min}} + \sum_{i \in [m]} B_1 \log^2\left(\frac{B_1 K}{\Delta_i^{\min}}\right) \log T\right); \tag{4}$$

*(2) if $\lambda = 1$, BCUCB-T (Algorithm 1) with an $(\alpha, \beta)$-approximation oracle achieves an $(\alpha, \beta)$-approximate regret bounded by*

$$O\left(\sum_{i \in [m]} \log\left(\frac{B_v K}{\Delta_i^{\min}}\right) \frac{B_v^2 \log K \log T}{\Delta_i^{\min}} + \sum_{i \in [m]} B_1 \log^2\left(\frac{B_1 K}{\Delta_i^{\min}}\right) \log T\right). \tag{5}$$

**Remark 5 (Discussion for Regret Bounds).** Looking at the above regret bounds, for $\lambda > 1$ and $\lambda = 1$, the leading terms are $O(\sum_{i=1}^{m} \frac{B_v^2 \log K \log T}{\Delta_i^{\min}})$ and $O(\sum_{i=1}^{m} (\log \frac{B_v K}{\Delta_i^{\min}}) \frac{B_v^2 \log K \log T}{\Delta_i^{\min}})$. When $B_v \geq B_1$ (which typically holds, see Section 5) and gaps are small (i.e., $\Delta_{\min}^i \leq 1/\log^2 K$), the dependencies over $K$ are $O(\log K)$ and $O(\log^2 K)$, respectively. For the setting of CMAB-T, [29] is the closest work to our paper, where the reward function satisfies Condition 1 and Condition 2 with coefficient $B_1$. As mentioned in Remark 3 in Section 3.1, their reward function trivially satisfies our Condition 3 with coefficient $(B_1 \sqrt{K}/2, B_1, 2)$ so our work reproduces a bound of $O(\sum_{i \in [m]} \frac{B_1^2 K \log K \log T}{\Delta_i^{\min}})$, matching [29] up to a factor of $O(\log K)$. As will be shown in Section 5, for applications that satisfy TPVM (or $\text{TPVM}_<$) condition with non-trivial $B_v$, i.e., $B_v = o(B_1 \sqrt{K})$, our work improves their regret bounds up to a factor of $O(K/\log K)$. As for the lower bound, according to the lower bound results by Merlis and Mannor [24], our regret bound is tight up to a factor of $O(\log^2 K)$ on the (degenerate) non-triggering CMAB case. We defer the details about the lower bound results and the distribution-independent regret bounds in the Appendix C.5.

*Proof ideas.* Our proof uses a few events to filter the total regret and then bound these event-filtered regrets separately. As will be shown in the supplementary material, the event that contributes to the leading regret is $E_t = \{\Delta_{S_t} \leq e_t(S_t)\}$, where the error term $e_t(S_t) = O(B_v \sqrt{\sum_{i \in \tilde{S}_t} (\frac{\log t}{T_{t-1,i}})(p_i^{D,S_t})^\lambda} + B_1 \sum_{i \in \tilde{S}_t} (\frac{\log t}{T_{t-1,i}})(p_i^{D,S_t}))$. To handle the probabilistic triggering, our key ingredient is to use the triggering probability group technique proposed by Wang and Chen [29] in the definition of above events. For the $\lambda = 1$ case, one new issue arises since the triggering probability group divides sub-optimal actions $S$ into *infinite* geometrically separated bins $(1/2, 1], (1/4, 1/2], ..., (2^{-j}, 2^{-j+1}), ...,$ over $p_i^{D,S}$, and the regret should be proportional to the number of bins (which are infinitely large). To handle this, we show that it suffices to consider the first $j \leq j_i^{\max} = O(\log \frac{B_v K}{\Delta_i^{\min}})$ bins (which is why Eq. (5) has this additional factor in the leading term) and the regret of other bins (with very small $p_i^{D,S}$) can be safely neglected. To bound the leading regret filtered by $E_t$ as mentioned earlier, we use the reverse amortization trick from Wang and Chen [29, 30] and adaptively allocates each arm's regret contribution (according to thresholds on the number of times arm $i$ is triggered). Note that these thresholds are carefully chosen for the error term $e_t(S_t)$, since trivially following the thresholds in Wang and Chen [29] would either yield no meaningful bound or suffer from additional $O(\log T)$ or $O(\log K)$ factors in the regret. As a by-product, one can also use our analysis to replace that of Merlis and Mannor [23] and Perrault et al. [25] (where similar error term $e_t(S_t)$ appears) to improve their bound by a factor of $O(\log K)$. For the detailed proofs, we defer them in the Appendix C. ∎

## 4 Algorithm and Analysis For CMAB with Independent Arms

In this section, we aim to show that for the non-triggering CMAB, the assumption that all arms are independent, compounded with a non-triggering version of the above TPVM condition (named as VM condition below), together allow us to completely remove the $O(\log^2 K)$ or $O(\log K)$ dependence

**Algorithm 2** SESCB: Sub-Exponential Sampling for Combinatorial Bandits with Independent Arms

1: **Input:** Base arms $[m]$, sub-Gaussian parameter $C_1$, VM smoothness coefficient $B_v$, $(\alpha, \beta)$-approximation ORACLE $\bar{O}$.
2: **Initialize:** For each arm $i$, $T_{0,i} \leftarrow 0, \hat{\mu}_{0,i} = 0$.
3: **for** $t = 1, ..., T$ **do**
4:     For all $S \in \mathcal{S}$, define min-count $T_{t-1,S}^{\min} = \min_{i \in S} T_{t-1,i}$, let interval $\rho_t(S) = B_v \sqrt{\sum_{i \in S} \frac{C_1}{T_{t-1,i}}} + \max\left\{ 8C_1 \sqrt{\sum_{i \in S} \frac{\log(2|\mathcal{S}|T)}{T_{t-1,i}^2}}, \frac{8C_1 \log(2|\mathcal{S}|T)}{T_{t-1,S}^{\min}} \right\}$.
5:     For all $S \in \mathcal{S}$, define optimistic reward $\bar{r}_t(S) = r(S; \hat{\boldsymbol{\mu}}_{t-1}) + \rho_t(S)$.
6:     Play $S_t = \bar{O}(\hat{\boldsymbol{\mu}}_t, \boldsymbol{T}_t)$ s.t. $\Pr\left[\bar{r}_t(S) \geq \alpha \cdot \bar{r}_t(\bar{S}_t^*)\right] \geq \beta$, where $\bar{S}_t^* = \arg\max_{S \in \mathcal{S}} \bar{r}_t(S)$, and observe outcome $X_{t,i}$'s, for $i \in S_t$.
7:     For every $i \in S_t$, update $T_{t,i} = T_{t-1,i} + 1, \hat{\mu}_{t,i} = \hat{\mu}_{t-1,i} + (X_{t,i} - \hat{\mu}_{t-1,i})/T_{t,i}$.
8: **end for**

in the existing regret bounds. In particular, we focus on the a non-triggering CMAB problem instance $([m], \mathcal{S}, \mathcal{D}, R)$. Its setting is similar to CMAB-T, but here we assume that $\mathcal{S}$ are collections of subsets of $[m]$ and only arms pulled by action $S_t \in \mathcal{S}$ are revealed as feedback (i.e., $\tau_t = S_t$).

**Condition 4** (VM Bounded Smoothness). *We say that a non-triggering CMAB problem instance $([m], \mathcal{S}, \mathcal{D}, R)$ satisfies the Variance Modulated (VM) $(B_v, B_1 \geq 0)$-bounded smoothness condition, if for any action $S \in \mathcal{S}$, any distribution $D, D' \in \mathcal{D}$ with mean vector $\boldsymbol{\mu}, \boldsymbol{\mu}' \in (0, 1)^m$, for any $\boldsymbol{\zeta}, \boldsymbol{\eta} \in [-1, 1]^m$ s.t. $\boldsymbol{\mu}' = \boldsymbol{\mu} + \boldsymbol{\zeta} + \boldsymbol{\eta}$, we have $|r(S; \boldsymbol{\mu}') - r(S; \boldsymbol{\mu})| \leq B_v \sqrt{\sum_{i \in S} \frac{\zeta_i^2}{(1-\mu_i)\mu_i}} + B_1 \sum_{i \in [m]} |\eta_i|$.*

**Condition 5** (Independent base arms). *We say that the base arms are independent, if for any $D \in \mathcal{D}$, the outcome vectors $\boldsymbol{X} \sim D$ are independent (across base arms), i.e., $D = \otimes_{i \in [m]} D_i$.*

**Condition 6** ($C_1 \mu_i (1 - \mu_i)$ sub-Gaussian). *The outcome distribution $D_i$ with mean $\mu_i$ is $C_1 \mu_i (1 - \mu_i)$ sub-Gaussian, where $C_1$ is a known coefficient.*

**Remark 6 (Comparison with TPVM Condition and [23]).** Condition 4 is the non-triggering version of TPVM, by setting $p_i^{D,S} = 1$ if $i \in S$ and 0 otherwise. As shown in Appendix B.2, Condition 4 can be implied by the original Gini-smoothness condition [23] with $(B_v, B_1) = (3\sqrt{2}\gamma_g, \gamma_\infty)$, so PMC application satisfies the VM condition (the fifth row in Table 3). But different from [23, Lemma 6] and TPVM$_<$, the VM condition is the undirectional version (i.e., we allow $\boldsymbol{\zeta}, \boldsymbol{\eta}$ to be negative). This is important for using empirical means in the algorithm (as we did in our SESCB policy), since they are not necessarily larger than the true means.

**Remark 7 (Motivation and Feasibility for Condition 6).** Condition 6 helps to cancel out the $(1 - \mu_i)\mu_i$ effect in the VM condition without explicitly using the empirical variance that will bring in additional batch-size dependent errors. For Bernoulli arms with mean $\mu_i$, we can compute the explicit value of $C_1$, i.e., $C_1 = \max_{i \in [m]} \frac{1 - 2\mu_i}{2 \ln(\frac{1-\mu_i}{\mu_i})(1-\mu_i)(\mu_i)}$ by [22]. Notice that $C_1$ could be large when $\mu_i$ is approaching 0 or 1, but it is safe to consider $\mu_i$ over bounded supports that are not too close to 0 or 1, e.g., when $\mu_i \in [0.01, 0.99]$, $C_1 \approx 10.78$.

**SESCB Algorithm.** Our proposed algorithm is shown in Algorithm 2. Instead of maintaining one upper confidence bound for each base arm $i$, we maintain an upper confidence bound for each super arm $S$, based on the estimated reward of the empirical means and a confidence interval. In line 4, we compute the confidence interval $\rho_t(S)$ by taking the max of two tentative segments within the square root, which corresponds to two different segments of the concentration bound for the sub-exponential random variable [27]. Such a sub-exponential concentrated confidence interval comes from the VM condition by treating $(\zeta_i)_{i \in S}$ as $|S|$ *independent* sub-Gaussian random variables, whose summation produces a more concentrated sub-exponential random variable compared with considering them as $|S|$ possibly dependent variables. It is notable that for the second tentative interval, SESCB uses the min-counter $T_{t-1,S}^{\min}$ instead of all counters $T_{t-1,i}$ in $S$, which is the key ingredient that removes the $O(\log K)$ factor as to be shown in the analysis. After getting $\rho_t(S)$, the optimistic reward is defined in Line 5 and the learner selects $S_t$ via the $(\alpha, \beta)$-approximation oracle $\bar{O}$ and updates the corresponding statistics.

**Regret Bound and Analysis.** The following theorem summarizes the regret bound for Algorithm 2.

**Theorem 2.** *For a non-triggering CMAB problem instance $([m], \mathcal{S}, \mathcal{D}, R)$ that satisfies VM bounded smoothness (Condition 4) with coefficient $(B_v, B_1)$, Condition 5 and Condition 6 with coefficient $C_1$,*

*SESCB (Algorithm 2) with an $(\alpha, \beta)$-approximation oralce achieves $(\alpha, \beta)$-approximate regret that is bounded by $O\left(\sum_{i\in[m]} \frac{B_v^2 \log T}{\Delta_i^{\min}} + \frac{B_v^2 mK}{\Delta_{\min}} + m\Delta_{\max}\right).$*

Looking at the above regret bound, the leading term totally removes the $O(\log K)$ dependency compared with Theorem 1. Compared with [23], our regret bounds improves theirs by $O(\log^2 K)$.

*Proof Ideas.* Similar to the proof of Theorem 1, we first identify an error term $e_t(S_t) = 2\rho_t(S_t)$ as Line 4 and consider the regret filtered by the event $\{\Delta_{S_t} \le e_t(S_t)\}$. The key ingredient is by following Condition 4 and Condition 6, and bound $|r(S; \hat{\boldsymbol{\mu}}) - r(S; \boldsymbol{\mu})| \le B_v\sqrt{\sum_{i\in S} u_{t,i}^2}$, where $u_{t,i}$ is a $(\frac{C_1}{T_{t-1,i}})$-sub-Gaussian random variable. Let $Y_{t,S} = \sum_{i\in S} u_{t,i}^2$. One can show $Y_{t,S}$ is a $(32C_1^2 \sum_{i\in S} \frac{1}{T_{t-1,i}^2}, 4C_1 \frac{1}{T_{t-1,S}^{\min}})$-sub-Exponential random variable, so applying the concentration bounds on $Y_{t,S}$ [27] and one can obtain the above $e_t(S_t)$. Then we consider two cases based on the value of $\sum_{i\in S_t} \frac{1}{T_{t-1,i}}$. For both cases, we use the reverse amortization trick from [29] but different from Section 3.2, $e_t(S_t)$ ensures that we only need to consider regret contributions from the min-arm (which is least played in $S_t$) according to certain batch-size independent thresholds. This in turn gives batch-size independent regret bounds that totally removes $O(\log K)$ in the leading term. See Appendix D for more details. ∎

**Computational Efficiency.** Notice that like other ESCB-type algorithms [9], for the general reward function $r(S; \boldsymbol{\mu})$, there may not exist efficient $\bar{O}$, so one needs to enumerate over all possible actions $S \in \mathcal{S}$ each round, where the time complexity could be as high as $O(|\mathcal{S}|T)$. However, when $r(S; \boldsymbol{\mu})$ is a monotone submodular function (e.g, the reward function of the PMC problem [8]), we can modify $\rho_t(S)$ so that the optimistic reward $\bar{r}_t(S)$ is also monotone submodular, which can be efficiently optimized with a greedy $(1 - 1/e, 1)$-approximation oracle. Observe that the current $\rho_t(S)$ is not submodular since the maximum of two submodular functions are not necessarily submodular, but we know the summation of two submodular functions are submodular. Based on this observation, we change $\rho_t(S)$ to $\rho_t'(S) = B_v\sqrt{\sum_{i\in S} \frac{C_1}{T_{t-1,i}}} + 8C_1\sqrt{\sum_{i\in S} \frac{\log(2|\mathcal{S}|T)}{T_{t-1,i}^2}} + \frac{8C_1 \log(2|\mathcal{S}|T)}{T_{t-1,S}^{\min}}$, where max is replaced with a sum $(+)$, and we prove in Appendix D.3 that $\rho_t'(S)$ is a monotone submodular function. Now we can use the greedy oracle to maximize a new optimistic reward $r_t'(S) = r(S; \hat{\boldsymbol{\mu}}_{t-1}) + \rho_t'(S)$ in our SESCB algorithm. As for the final regret, using $\rho_t'(S)$ instead of $\rho_t(S)$ only worsens the final regret by a constant factor of two.

Now compared with [23] that achieves $(1 - 1/e, 1)$-approximate regret bound for PMC problem, our SESCB achieves the same $(1 - 1/e, 1)$-approximate regret bound but completely removes the $O(\log K)$ dependency. Moreover, our greedy oracle is efficient with computational complexity $O(TKL)$, where $T$ is the total number of rounds, $K$ is the number of source nodes to be selected in each round and $L$ is the total number of source nodes, which is much faster than the enumeration method. For the regret analysis when using $r_t'(S)$, see Appendix D.3 for more details.

## 5 Applications

In this section, we show how various applications satisfy our new TPVM, TPVM$_<$ or VM smoothness condition and their corresponding $(B_v, B_1, \lambda)$ coefficients with non-trivial $B_v$, i.e., $B_v = o(B_1\sqrt{K})$, which in turn improves the regret bounds over the batch-size dependence of $K$.

**Theorem 3.** *The combinatorial cascading bandits [17], the multi-layered network exploration [21], the influence maximization problems [29] and the probabilistic maximum coverage problem [23] satisfy the TPVM (TPVM$_<$ or VM) conditions with coefficients $(B_v, B_1, \lambda)$, resulting regret bounds and improvements shown in Table 3.*

Note that the first four applications in Table 3 applies Theorem 1, while the last application applies Theorem 2. More specifically, the first two applications we consider are disjunctive and conjunctive cascading bandits [17], where $m$ base arms represent web pages and routing edges in online advertising and network routing, respectively. Batch-size $K$ is the maximum size of the ordered sequence $S \in \mathcal{S}$ to be selected in each round, which will trigger web pages/routing edge one by one until certain stopping condition is satisfied, i.e., a click or a routing edge being broken. The reward is 1

Table 3: Summary of the coefficients, regret bounds and improvements for various applications.

| Application | Condition | $(B_v, B_1, \lambda)$ | Regret | Improvement |
|---|---|---|---|---|
| Disjunctive Cascading Bandits [17] | TPVM$_<$ | $(1,1,2)$ | $O(\sum_{i\in[m]} \frac{\log K \log T}{\Delta^{\min}})$ | $O(K/\log K)$ |
| Conjunctive Cascading Bandits [17] | TPVM | $(1,1,1)$ | $O(\sum_{i\in[m]} \log \frac{K}{\Delta_{\min}} \frac{\log K \log T}{\Delta_{i,\min}})$ | $O(K/(\log K \log \frac{K}{\Delta_{\min}}))$ |
| Multi-layered Network Exploration [21] | TPVM | $(\sqrt{1.25|V|},1,2)^\dagger$ | $O(\sum_{i\in\mathcal{A}} \frac{|V|\log(n|V|)\log T}{\Delta^{\min}})$ | $O(n/\log(n|V|))$ |
| Influence Maximization on DAG [29] | TPVM | $(\sqrt{L}|V|,|V|,1)^\dagger$ | $O(\sum_{i\in[m]} \log \frac{|E|}{\Delta_{\min}} \frac{L|V|^2\log|E|\log T}{\Delta^{\min}})$ | $O(|E|/(L\log|E|\log \frac{|E|}{\Delta_{\min}}))$ |
| Probabilistic Maximum Coverage [23]$^*$ | VM | $(3\sqrt{2|V|},1,-)$ | $O(\sum_{i\in[m]} \frac{|V|\log T}{\Delta^{\min}})$ | $O(\log^2 k)$. |

$^*$ This row is for the application in Section 4 and the rest of rows are for Section 3.1; $^\dagger$ $|V|,|E|,n,k,L$ denotes the number of target nodes, the number of edges that can be triggered by the set of seed nodes, the number of layers, the number of seed nodes and the length of the longest directed path, respectively.

if *any* web page is clicked (or if *all* routing edges are live) and 0 otherwise. Compared with [29], we achieve an improvement $O(K/(\log K \log \frac{K}{\Delta_{\min}}))$ for the conjunctive case and an improvement $O(K/\log K)$ for the disjunctive case, due to the same $B_v, B_1 = 1$ but different orders $\lambda$.

The third application is the mutli-layered network exploration (MuLaNE) problem [21], and the MuLaNE task is to allocate $B$ budgets into $n$ layers to explore target nodes $V$. In MuLaNE, the base arms form a set $\mathcal{A} = \{(i,u,b) : i \in [n], u \in [V], b \in [B]\}$, the batch-size $K = (n+1)|V|$ and the reward is defined as the total reward give by the first visit of any target nodes. MuLaNE fits into our study, and compared with [21], the regret bound is improved by a factor of $O(n/\log K)$.

Our fourth application is the online influence maximization (OIM) problems direct acyclic graphs (DAG). For this application, the goal is to select at most $k$ seed nodes to influence as many target nodes $V$ as possible, where the influence process follows the independent cascade (IC) model [29] (see Appendix E for more details). The base arms are the edges with unknown edge probabilities and the batch-size $K$ is the total number of edges that could be triggered by any set of $k$ seed nodes, denoted as $|E|$. The improvements here are significant, improving the existing results [23] by a factor of $O(|E|/(L\log|E|\log \frac{|E|}{\Delta_{\min}}))$.

For the PMC problem [23], we consider a complete bipartite graph with $L$ source nodes on the left and $|V|$ target nodes $V$ on the right. The goal is to select $k$ seed nodes from $L$ nodes trying to influence as many as target nodes, so the edges $E$ are independent base arms and the batch-size is $k|V|$. By using the computational efficient version of Algorithm 2 and applying Theorem 2, we achieve $O(\log^2 k)$ improvement compared with [23] while maintaining good computational efficiency.

*Proof Ideas.* For all above applications (except for the OIM on DAG), our proof involves the use of telescoping series to decompose the reward difference, together with a smart use of the Cauchy–Schwarz inequality aided by the variance terms. For disjunctive cascading bandits, for example, the reward difference $|r(S;\bar{\boldsymbol{\mu}}) - r(S;\boldsymbol{\mu})| = \prod_{i=1}^{K}(1-\mu_i) - \prod_{i=1}^{K}(1-\bar{\mu}_i)$ can be telescoped as $\sum_{i\in[K]}(\bar{\mu}_i - \mu_i)\left(\prod_{j=1}^{i-1}(1-\mu_j) \cdot \prod_{j=i+1}^{K}(1-\bar{\mu}_j)\right)$. After this decomposition, we replace certain terms with $p_i^{D,S}$ and bound above by $\sum_{i\in[K]} \zeta_i p_i^{D,S} \sqrt{\prod_{j=i+1}^{K}(1-\mu_j)} + \sum_{i\in[K]} \eta_i p_i^{D,S}$. Then we simultaneously multiply and divide the variance term $\sqrt{(1-\mu_i)\mu_i}$ on the first term and apply the Cauchy–Schwarz inequality to move the summation over $K$ into the square root, concluding the satisfaction of Condition 3 with $B_v = \sqrt{\sum_{i\in[K]}(1-\mu_i)\mu_i \prod_{j=i+1}^{K}(1-\mu_j)} \le 1$. As for the OIM on DAG, since reward function have no closed-form solutions [5], the analysis is more involved with the need of advanced techniques such as the coupling technique [20], see Appendix E for details. ∎

## 6   Conclusion and Future Direction

This paper studies the CMAB problem with probabilistically triggered arms or independent arms. We discover new TPVM and VM conditions, and propose BCUCB-T and SESCB algorithms to reduce and remove the batch-size $K$ in the regret bounds, respectively. We also show that several important applications all satisfy our conditions to achieve improved regrets, both theoretically and empirically. There are many compelling directions for future study. For example, it would be interesting to study the setting of CMAB-T together with independent arms. One could also explore how to extend our application and consider general graphs in online influence maximization bandits.

## Acknowledgments and Disclosure of Funding

The work of John C.S. Lui was supported in part by the HK RGC SRF2122-4202. The work of Siwei Wang was supported in part by the National Natural Science Foundation of China Grant 62106122.

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
