# OpenReview forum: "Batch-Size Independent Regret Bounds for Combinatorial Semi-Bandits with Probabilistically Triggered Arms or Independent Arms"
_NeurIPS.cc/2022/Conference — NeurIPS 2022 Accept_

### Official Review · Reviewer_kHgZ · 2022-07-07

**Rating:** 7
**Confidence:** 4
**Soundness:** 3 good
**Presentation:** 2 fair
**Contribution:** 3 good

**Summary:**

Remark: Throughout my review, [n] refers to the n-th reference in the full paper (with appendix) from the supplementary material. (I mention this because the reference numbers differ in the 9-page submission.)

This paper studies Combinatorial Multi-Armed Bandit (CMAB) problems with probabilistically-Triggered arms (CMAB-T). In essence, CMAB is a variant of the standard bandit setting where the learner chooses a subset of arms (a.k.a. a “super arm” or “action”) and the mean reward is a function of the chosen subset and of its component arms’ mean rewards, and for CMAB-T the learner only observes feedback on a random subset of the chosen arms. This formulation generalizes problems including cascading bandits for ranking search results, online influence maximization, etc.

Prior work [27] introduced a smoothness condition (with respect to the function that maps the component arms’ mean rewards to the subset’s mean reward) that improved existing regret bounds [7] by a factor of $1/p^*$ (see Line 36 for details). This work provides a refined smoothness condition that involves the variance of the arms’ reward distribution, i.e., the condition is less restrictive when the variances are smaller (simply because the mean rewards are easier to learn in such cases). Provided this condition holds, the authors develop a variance-aware UCB-style algorithm based on the empirical Bernstein inequality [1] and show its regret improves existing work with respect to the “batch size” K, which is the maximum of arms that can be triggered at each round (e.g., the number of search results that a cascading bandit algorithm chooses).

In addition, the authors study non-triggered arms (Section 4) and specialize their results to various application settings (Section 5).

**Questions:**

~~In the first row of Table 3, the authors claim that their regret bound for disjunctive cascading bandits improves the existing one from [16] by a factor of $K / log K$. The existing bound is not explicitly shown, but the authors’ bound has the form $log(K) log(T) \sum_i \Delta_i^{-1}$, so I assume the existing bound refers to the bound $K \log(T) \sum_i \Delta_i^{-1}$ from [16]. However, as best I can tell, this bound from [16] is specialized to cascading bandits from a more general setting; if one restricts to the special case like in the original cascading bandit paper [15], the bound $\log(T) \sum_i \Delta_i^{-1}$ from Theorems 2 and 3 in [15] actually seems better than the bound from the current paper ... or at least, much better than the bound in [16], since [15] has no explicit multiplicative dependence on $K$? (Note: [15] does not refer to its model as "disjunctive", but I believe it's the same formulation.)~~

Update post-rebuttal: This question has been satisfactorily addressed and I remain in support of acceptance.

**Limitations:**

Overall, I feel the limitations were acknowledged -- for example, the assumptions are clearly stated in each of the theorems, Remark 3 acknowledges that additional assumptions are needed to ensure the proposed Condition 3 implies the existing Condition 2, etc. In terms of negative societal impact, the authors simply state (in the checklist) that there is no foreseeable impact. Personally, I feel this view is somewhat narrow -- while the paper is theoretical, the algorithms could obviously have real-world impact if deployed -- but I understand the authors' point and view this more as a difference of opinion than an objective weakness.

**Strengths And Weaknesses:**

STRENGTHS:

In my opinion, the strengths of the paper are (1) a novel and fairly natural smoothness condition that incorporates variance information, (2) algorithms that exploit low variance problem instances via empirical Bernstein confidence sets (instead of “variance-unaware” Hoeffding-based confidence sets), and (3) analysis that shows exploiting variance in this manner can dramatically reduce regret in terms of $K$ (e.g., from $K$ to $log K$ in some settings).

At a higher level, I feel the main strength of this paper is to show that variance-aware algorithms lead to polynomial improvement in terms of K, and to do so in a fairly general setting (CMAB-T). More specifically, paper A (see below) recently proved something similar for cascading bandits (a special case of CMAB-T), although it focuses on gap-free bounds so is complementary to the current work (which focuses on gap-dependent bounds). More broadly, a number of recent papers have demonstrated similar (in spirit) polynomial improvements resulting from variance-aware algorithms in various bandit and RL settings (e.g., paper C below achieves a polynomial improvement in terms of the horizon for finite-horizon RL; see also the references therein and in paper B below). Thus, the current paper seems rather timely given the similar flavor to these works.

(To be clear, I’m not demanding these papers be cited — reference A is a very recent preprint and the others are on different topics — but rather, emphasizing the timeliness of the current work).

(A) Vial, Sanghavi, Shakkottai, Srikant, “Minimax Regret for Cascading Bandits”, arXiv preprint

(B) Zhang, Yang, Ji, Du, “Improved Variance-Aware Confidence Sets for Linear Bandits and Linear Mixture MDP”, arXiv preprint

(C) Zhou, Gu, Szepesvari, “Nearly Minimax Optimal Reinforcement Learning for Linear Mixture Markov Decision Processes”, COLT 2021

WEAKNESSES:

(1) While mostly well-written, the paper is very dense, particularly the first few technical sections (i.e., starting from Section 2). I imagine this is mostly a side effect of packing many technical definitions and results into a short page limit, but it was hard to follow at times. For example, Section 2 would have been easier to understand if the authors had used a simple special case of the CMAB-T model (e.g., cascading bandits) as a running example to help illustrate the technical definitions (of which there are many in Section 2).

(2) Along these lines, I did not find the proof sketches very illuminating, and it was difficult to glean much intuition from the full proofs in the appendix given their length and density. In other words, I would have preferred an intermediate explanation -- high-level like the proof sketches, but more detailed like the actual proofs -- to illustrate the key ideas of the analysis (again, perhaps in a simpler special case of CMAB-T).

~~(3) Unless I'm mistaken (in which case I'll gladly update during the discussion period), the improvement over prior work in the disjunctive cascading bandit setting is oversold. (See "Questions" section for details.)~~

Update post-rebuttal: Weakness (3) has been satisfactorily addressed and I remain in support of acceptance.

---

> ### Author Response · Authors · 2022-08-01
> **About the related works and the presentation.**
>
> We thank the reviewer for mentioning three related works about variance-aware algorithms as well as the kind suggestion on our writing. For the related works, (A) is an interesting concurrent work that studies cascading bandits, which shares a similar variance-aware principle as ours. (A) studies two settings: the tabular case which is overlapped with us (though our work focuses on a slightly more general combinatorial cascading bandits [16], see the discussion above this reply) and the linear contextual cascading bandit case. Compared with the overlapped tabular case, our BCUCB-T achieves the matching regret bound when translating to gap-independent regret bound as in Appendix C.5.3. Interestingly, (A) also discusses the regret lower bound, which can show the tightness of our results as well. Moreover, our work is on a more general framework for CMAB-T, summarized by the TPVM condition, while (A) only focuses on cascading bandits.
> For the linear contextual case studied by (A), we do not consider this setting in our work and it will be an interesting future direction to see whether the TPVM condition can also be applied to handle the linear contextual CMAB. For related works (B) and (C), we will add them to emphasize the timeliness of the current work as suggested. For the writing, we will add more examples (e.g., combinatorial cascading bandits) in the proofs and discussions to illustrate the intuitions of our definitions, assumptions, and proof techniques, which would be helpful for possible follow-up works.

---

> ### Author Response · Authors · 2022-08-01
> **About the improvement over prior work in the disjunctive cascading bandit setting.**
>
> We thank the reviewer for the positive comments. We agree with the reviewer that our regret does in fact have an extra $\log K$ term compared with [15], and we apologize for the confusion on this point. However, we believe it is reasonable to compare with [16] whose regret bound is $O(K\sum_{i}\frac{\log T}{\Delta_{i}})$, rather than with [15] whose regret bound is $O(\sum_{i}\frac{\log T}{\Delta_{i}})$. The reason is as follows.
>
> The main difference between these two works lies in the set of feasible actions  $\mathcal{S}$.
> For [15], $\mathcal{S}$ is the collection of all permutations whose size equals to $K$ (i.e., a uniform matroid). In this case, the items in the feasible solutions are **exchangeable** (a critical property for matroids), i.e., $S - \{e_1\} + \{e_2\} \in \mathcal{S}$, for any $S \in \mathcal{S}, e_1, e_2 \in [m]$. Based on the exchangeability property, [15] is able to define the item-wise sub-optimal gap $\Delta_{e,e^*}$ in Eq. (3) of [15] and the critical event $G_{e,e^*,t}$ in Theorem 1 of [15]. With these definitions, their proof is basically bounding the number of times that each suboptimal item $e$ is chosen instead of any optimal item $e^*$, which yields a batch-size independent regret bound $O(\sum_{i}\frac{\log T}{\Delta_{i}})$.
>
> For [16], however, $\mathcal{S}$ (i.e., $\Theta$ in [16]) consists of arbitrary feasible actions (perhaps with different sizes), e.g., $S \in \mathcal{S}$ could refer to any path that connects the source and the destination in network routing applications.
> [16] refers to cascading bandits with this kind of $\mathcal{S}$  as "combinatorial cascading bandits", where items in the feasible actions may not be exchangeable (e.g., deleting an edge from a path and adding another edge may not be a valid path anymore). This means the item-wise definition of $\Delta_{e,e^*}$ and $G_{e,e^*, t}$ no longer works. Instead of using the $\Delta_{e,e^*}$, [16] uses another definition for the sub-optimal gap $\Delta_{e,\min}$ that is similar to our Definition 1, and applies a more general proof based on [17] that yields an extra $K$ factor.
>
> Similar to [7], [16], and [17], our CMAB-T formulation focuses on the arbitrary $\mathcal{S}$. In other words, our BCUCB-T algorithm and its analysis can deal with this general $\mathcal{S}$ whose feasible actions are not exchangeable. So we believe that we should compare [16] with our result, and it is unfair to compare ours with [15] that explicitly assumes $\mathcal{S}$ to be the uniform matroid, which enjoys the additional "exchangeable property".
> We will add this discussion in our final version to make it clear.

---

> > ### Comment · Reviewer_kHgZ · 2022-08-03
> > **Thanks for the clarification**
> >
> > Thanks for the clarification. I read your detailed response and revisited the relevant papers. This addressed my concern and I updated my review accordingly.

---

### Official Review · Reviewer_uueH · 2022-07-10

**Rating:** 7
**Confidence:** 4
**Soundness:** 4 excellent
**Presentation:** 4 excellent
**Contribution:** 3 good

**Summary:**

This paper proposes new smoothness conditions for the well-known combinatorial semi-bandit problem with probabilistically triggered arms. It is shown that well-known combinatorial learning problems of interest, including combinatorial cascading, influence maximization, and probabilistic maximum coverage bandits, satisfy these conditions. With the so-called triggering probability and variance modulated smoothness condition, the authors significantly improved the O(K) factor that appears in the previous regret bounds, where K represents the batch size. The authors also propose a non-triggering version of the smoothness condition and show that dependency on K can be completely removed for the non-triggering combinatorial semi-bandit. In order to achieve the improved regret bounds for the triggering version of the problem, the paper relies on empirical Bernstein inequality to construct upper confidence bounds of base arms. For the non-triggering version, batch size independence is achieved by constructing sub-exponential concentrated confidence intervals.

**Questions:**

I wonder if any computationally efficient algorithm can achieve batch size independence.

**Limitations:**

Limitations are adequately discussed. This work is theoretical in nature and does not have a societal impact.

**Strengths And Weaknesses:**

Strengths:

Finding general enough smoothness conditions under which the effect of batch size on the regret can be significantly improved is an important problem. This paper intuitively develops new smoothness conditions and algorithms that use tailored confidence intervals to address this problem.

Computationally efficient algorithms are proposed for triggering and non-triggering cases.

Writing is clear. Theorems are well organized.

Weakness:

True batch size independence comes with an algorithm that requires enumeration of all possible actions. I wonder if any computationally efficient algorithm can achieve batch size independence.

---

> ### Author Response · Authors · 2022-08-01
> **About the computationally efficient algorithm for SESCB (Algorithm 2).**
>
> We thank the reviewer for the positive comments. The reviewer raises a concern about the computational efficiency of our SESCB algorithm, which can completely remove the $O(\log K)$ dependence. As we mentioned in line 318, the computation efficiency issue comes from the fact that the SESCB enumerates all possible actions, which is the same issue experienced by other ESCB-type algorithms that also use the brute force method, e.g., [8], [24].
>
> One way to avoid the brute-force search is to introduce an $(\alpha,\beta)$-approximation oracle that could be efficient, as mentioned in our reply to the first question of reviewer HNeW.
> Generally speaking, we can define $\bar{r}\_t(S)$ for all $S \in \mathcal{S}$ instead of explicitly computing them. We then treat $\bar{r}\_t(S)$ as a general set function. Now can we assume an $(\alpha, \beta)$-approximation oracle $\bar{O}$ which can efficiently produce $S \in \mathcal{S}$ such that $\Pr\left[\bar{r}\_t(S)\ge \alpha \cdot \bar{r}\_t(\bar{S}^\*\_t)\right] \ge \beta$, where $\bar{S}^*\_t=\arg\max\_{S\in \mathcal{S}} \bar{r}\_t(S)$. In this way, we can easily change the regret to $(\alpha,\beta)$-approximate regret to trade-off the efficiency.
>
> For the existence of such an oracle, the main difficulty lies in whether one can efficiently solve the optimization problem over a non-linear set function $r(S;\hat{\boldsymbol{\mu}}\_{t-1})$ plus another non-linear set function $\rho\_t(S)$.
> In fact, we can find an efficient $(1-1/e, 1)$ greedy oracle when the reward function $r(S;\hat{\boldsymbol{\mu}}\_{t-1})$ and $\rho\_t(S)$ are both monotone submodular functions.
>
> Take the PMC problem for example, for which $r(S;\boldsymbol{\mu})$ is monotone submodular. To make the interval $\rho\_t(S)=B\_v\sqrt{\sum\_{i \in S} \frac{C\_1}{T\_{t-1,i}}+ \max\left\\{8C\_1\sqrt{\sum\_{i \in S}\frac{\log(2|\mathcal{S}|T)}{T\_{t-1,i}^2}}, \frac{8C\_1\log(2|\mathcal{S}|T)}{T^{\min}\_{t-1, S}}\right\\}}$ submodular, we can change it to $\rho'\_t(S)=B\_v\sqrt{\sum\_{i \in S} \frac{C\_1}{T\_{t-1,i}}+ 8C\_1\sqrt{\sum\_{i \in S}\frac{\log(2|\mathcal{S}|T)}{T\_{t-1,i}^2}}+ \frac{8C\_1\log(2|\mathcal{S}|T)}{T^{\min}\_{t-1, S}} }$, where the $\max$ is replaced with a sum ($+$). We know that $g(f(S))$ is submodular if $f(S)$ is submodular and $g$ is a non-decreasing concave function, so it suffices to show three terms within the (non-decreasing concave) square root  in $\rho'\_t(S)$ are submodular. The first term is a modular function, the second term is the square root of a modular function, and the third term can be rewritten as $\max\_{i \in S}\frac{8C\_1\log(2|\mathcal{S}|T)}{T\_{t-1, i}}$, which is also monotone submodular. Now we can use the greedy oracle to maximize a new optimistic reward $\bar{r}\_t(S)=r(S;\hat{\boldsymbol{\mu}}\_{t-1})+\rho'\_t(S)$ in our SESCB algorithm. As for the final regret, using $\rho'\_t(S)$ instead of $\rho\_t(S)$ only worsens the final regret by a constant factor, since it only affects the analysis of case 1 of Appendix D.2 by multiplying a factor of two in line 1006 to deal with the larger $\rho'\_t$. Now compared with Merlis and Mannor [22] that achieves $(1-1/e, 1)$-approximate regret bound, our SESCB achieves the same $(1-1/e, 1)$-approximate regret bound but completely removes the $O(\log K)$ dependency. Moreover, our greedy oracle is efficient with computational complexity $O(TKL)$, where $T$ is the total number of rounds, $K$ is the number of source nodes to be selected in each round and $L$ is the total number of source nodes, which is much faster than our previous enumeration method. We will add the above discussion to improve the computational efficiency of SESCB.

---

> > ### Comment · Reviewer_uueH · 2022-08-07
> > **Thank you for the response**
> >
> > Thank you for the response. This will be a good addition to the final version of the paper.

---

### Official Review · Reviewer_HNeW · 2022-07-16

**Rating:** 6
**Confidence:** 3
**Soundness:** 3 good
**Presentation:** 4 excellent
**Contribution:** 3 good

**Summary:**

This paper considers the combinatorial semi-bandits problem under two different settings: (1) probabilistically triggered arms setting where the set of played arms can trigger reward on other arms, (2) the non-triggering setting with independent arms. For both these settings the authors improve batch-size dependence in regret as compared to previous work by considering a new variance-based condition on the underlying reward distributions. The authors show that this condition is satisfied in many applications such as cascading bandits, influence maximization on DAGs etc, and leads to significant improvements in the regret achievable for these applications.

**Questions:**

1. In Section 4, is the notion of regret still (\alpha, \beta)-approximate regret? If yes, then I do not understand the reason behind using this notion of regret. In the independent arms setting you are not using the oracle to return a (\alpha, \beta)-approximate set of arms, instead you are computing the argmax using brute force. In that case you should be able to compete with the best set.

2. It would be great if the authors can provide experimental results for some applications which support the theoretical improvements.

**Limitations:**

I do not foresee any negative societal impact.

**Strengths And Weaknesses:**

Strengths: The paper provides a good contribution to the literature on combinatorial semi-bandits as it improves the dependence on batch size from linear to logarithmic in many practical applications. The paper is very well-written and provides clear intuition behind various ideas/assumptions. The comparison with prior work is also adequate.

---

> ### Author Response · Authors · 2022-08-01
> **About the experimental results.**
>
> We thank the reviewer for this question. We have provided experiments to support the theoretical improvements of our work in the Appendix F. Specifically, we consider two representative applications, the combinatorial cascading bandits (Appendix F.1) and the PMC bandit (Appendix F.2), for the CMAB-T and the CMAB with independent arms, respectively. Compared with benchmark algorithms, our BCUCB-T algorithm achieves about 20\% less regrets for the combinatorial cascading bandits, and our SESCB achieves about 15\% less regrets for the PMC bandits. We will move these experimental results into our main paper in a future version.

---

> ### Author Response · Authors · 2022-08-01
> **Is the notion of regret still $(\alpha, \beta)$-approximate regret in Section 4?**
>
> We thank the reviewer for raising this good question. The regret in Section 4 is the exact regret, i.e., $\alpha=\beta=1$, since our SESCB algorithm (Algorithm 2) enumerates all possible solutions and selects the optimal one.
>
> Inspired by the reviewer's question, we can actually generalize our SESCB algorithm by allowing an $(\alpha, \beta)$-approximation oracle, similar to line 5 of our BCUCB-T algorithm. In particular, we can change lines 4-7 by defining $\bar{r}\_t(S)$ for all $S \in \mathcal{S}$ instead of explicitly computing them. We then treat $\bar{r}\_t(S)$ as a general set function, described by $2m$ parameters consisting of $m$ empirical means $\hat{\mu}\_{t-1,i}$ and $m$ counters $T\_{t,i}$, for $i \in [m]$. Now we assume an $(\alpha, \beta)$-approximation oracle $\bar{O}:[0,1]^m \times \mathbb{Z}^{m}\rightarrow \mathcal{S}$ which can produce
> \begin{align}
>    S=\bar{O}(\hat{\boldsymbol{\mu}}\_{t-1}, \boldsymbol{T}\_{t}) \text{ s.t. }  \text{Pr}\left[\bar{r}\_t(S)\ge \alpha \cdot \bar{r}\_t(\bar{S}^\*\_t)\right] \ge \beta,
> \end{align}
> where $\bar{S}^\*\_t=\arg\max\_{S\in \mathcal{S}} \bar{r}\_t(S)$.
> In this case, we can improve the computational efficiency of SESCB when there exists an efficient oracle that can
> 	(approximately) optimize the set function $\bar{r}\_t(S)$ over $S \in \mathcal{S}$. For the analysis, it is straightforward to change Eq. (136)-(139) as follows,
> \begin{align}
> \Delta\_S &= \alpha r(S^*; \boldsymbol{\mu}) - r(S;\boldsymbol{\mu})\\\\
> &\le \alpha ( r(S^*;\hat{\boldsymbol{\mu}}\_{t-1}) + \rho\_t(S^*)) - r(S; \boldsymbol{\mu}) \\\\
> &\le \bar{r}\_t(S) - r(S; \boldsymbol{\mu})\\\\
> &= r(S;\hat{\boldsymbol{\mu}}\_{t-1}) + \rho_t(S) - r(S; \boldsymbol{\mu})  \\\\
> &\le 2\rho\_t(S),
> \end{align}
> where the first inequality is because of Eq. (135) over $S^*$, the second inequality is due to the $(\alpha, \beta)$-approximation oracle $\bar{O}$ mentioned above, and the last inequality is due to Eq. (135) over $S$.
> Since the last inequality remains exactly the same as Eq. (139) in the previous version, this change does not affect later analysis. As for the $\beta$ part, we can apply the similar proof as in Eq. (54) to bound the regret when the oracle fails by $(1-\beta)T\Delta_{\max}$, which is absorbed by our $(\alpha, \beta)$-approximate regret definition, and the final regret bound remains unchanged. The only difference is that we have an $(\alpha, \beta)$-approximate regret, instead of the exact regret given by the enumeration.
>
> For the existence of such efficient oracles, we can give a concrete example on the PMC problem. Specifically, we are able to modify $\bar{r}\_t(S)$ in Algorithm 2 so that it can be proved to be a submodular function. We then use the submodular maximization technique to find an approximate solution to $\bar{r}\_t(S)$ , which is essentially a greedy $(1-1/e,1)$-approximation oracle. Solving this problem is quite efficient and achieves the state-of-the-art $(1-1/e,1)$-approximation regret bound for PMC bandit, removing the $O(\log K)$ factor. For more details, see our response to Reviewer uueH.
>
> We will add the above clarification and discussion in the final version to make it clear.

---

> ### Author Response · Authors · 2022-08-10
> **Any further questions?**
>
> Dear reviewer,
>
> We wonder if our response has addressed your question about the $(\alpha,\beta)$-approximation regret and the experiments. We are happy to have a further discussion if you have more questions.

---

### Meta-Review · Area_Chair_YNVU · 2022-08-20

**Recommendation:** Accept
**Confidence:** Certain

**Metareview:**

Thank the authors for their submission.

The paper studies combinatorial multi-armed bandit with probabilistically triggered arms. That is an MAB setting in which, at each round, the learner chooses a subset of the arms and obtains a reward that is some function of expected rewards of the chosen arms. In addition, the learner only observes feedback on a random subset of her chosen arms (triggered arms).

The paper relaxes a smoothness assumption laid by a previous work, and further improves the dependence on K in the regret bound, where K is the batch size (maximum number of triggered arms)
The authors provide computationally-efficient algorithms that are based on Bernstein concentration inequality, facilitating the improved bounds.
The paper is well-written and organized, and the theoretical results are sound.

**Award:**

No

---

### Decision · Program_Chairs · 2022-09-14

Accept